# Epigenetic Changes as a Target in Aging Haematopoietic Stem Cells and Age-Related Malignancies

**DOI:** 10.3390/cells8080868

**Published:** 2019-08-10

**Authors:** Sonja C. Buisman, Gerald de Haan

**Affiliations:** European Research Institute for the Biology of Ageing, University Medical Center Groningen, University of Groningen, 9700 Groningen, The Netherlands

**Keywords:** aging, haematopoietic stem cells, epigenetics, therapeutic targeting, age-related haematopoietic malignancies

## Abstract

Aging is associated with multiple molecular and functional changes in haematopoietic cells. Most notably, the self-renewal and differentiation potential of hematopoietic stem cells (HSCs) are compromised, resulting in myeloid skewing, reduced output of red blood cells and decreased generation of immune cells. These changes result in anaemia, increased susceptibility for infections and higher prevalence of haematopoietic malignancies. In HSCs, age-associated global epigenetic changes have been identified. These epigenetic alterations in aged HSCs can occur randomly (epigenetic drift) or are the result of somatic mutations in genes encoding for epigenetic proteins. Mutations in loci that encode epigenetic modifiers occur frequently in patients with haematological malignancies, but also in healthy elderly individuals at risk to develop these. It may be possible to pharmacologically intervene in the aberrant epigenetic program of derailed HSCs to enforce normal haematopoiesis or treat age-related haematopoietic diseases. Over the past decade our molecular understanding of epigenetic regulation has rapidly increased and drugs targeting epigenetic modifications are increasingly part of treatment protocols. The reversibility of epigenetic modifications renders these targets for novel therapeutics. In this review we provide an overview of epigenetic changes that occur in aging HSCs and age-related malignancies and discuss related epigenetic drugs.

## 1. Epigenetic Regulation of HSCs

Haematopoiesis is a tightly regulated process and involves the control of transcription of genes that guard the balance between self-renewal of HSCs and their proper differentiation into all mature blood cell lineages. As primitive stem cells differentiate towards functionally mature peripheral blood cells, massive genome-wide changes in gene expression patterns occur [1]. In contrast, when HSCs self-renew at least one of the two daughter cells is expected to inherit a gene expression pattern that is highly similar, if not identical, to the mother stem cell. Genes involved in differentiation pathways must be repressed in HSCs, while self-renewal genes are repressed upon differentiation. Complex epigenetic control mechanisms play a crucial role in coordinating gene expression patterns that guard this balance between self-renewal and differentiation. Such epigenetic modifications involve both DNA methylation and chromatin conformation and, thus, impact the accessibility of DNA to transcription factors and the transcriptional machinery. Posttranslational modifications of histones that affect chromatin conformation mostly include methylation, acetylation and ubiquitination of histone tails (Figure 1).

Such epigenetic modifications can be actively deposited or removed by epigenetic enzymes. The role of these enzymes in haematopoiesis and leukemogenesis will be the main focus of this review. Passive (non-enzyme mediated) changes of epigenetic modifications also take place and are part of the replication process of cells. Every cell division changes in epigenetic landscape occur, most of which are temporary. For example, during DNA replication the newly formed strand is generated without the methylation marks present on the template strand, referred to as passive DNA methylation. In contrast to active DNA demethylation, DNA replication associated passive demethylation is not complete, since methylation marks on the parental strand remain. The parental methylation pattern on the newly formed strand is deposited by DNA methylases, such as DNMT1 [2]. Beyond DNA methylation, the parental epigenetic histone marks also have to be established on the newly synthesized histones. Replication stress can disturb posttranscriptional deposition of the parental chromatin marks and DNA methylation state and, thus, lead to epigenetic changes [3].

On the other hand, new DNA methylation patterns and histone modifications are actively deposited or removed by dedicated enzymes, or recognized by specific readers. Importantly, a large body of evidence has demonstrated an important, if not essential, role for many of these epigenetic writers, readers, and erasers in normal haematopoiesis. Multiple epigenetic proteins have been associated to control self-renewal capacity of HSCs. Table 1 provides an overview of the most prominent of such epigenetic haematopoietic stem cell genes. 

## 2. Epigenetic Changes during Normal Aging of Murine HSCs

Genome-wide changes of epigenetic modifications are not only observed upon differentiation, but have also been identified during normal HSC aging. Studies in mice have shown that aged HSCs have low self-renewal potential [69,70,71,72,73], which appears to be inversely related to the number of cell divisions a HSC has undergone [74]. Importantly, the functional capacity of individual murine HSCs is highly variable and this variation increases during aging. Whereas the large majority of aged HSCs performs worse than their young counterparts, there are also aged HSC that function just fine [72,75,76]. This massive cell-to-cell variability is unlikely to result from the accumulation of random genetic mutations, as HSCs divide only very rarely during the lifespan of an organism [77]. Instead, functional heterogeneity is easier explained by random alterations of epigenetic modifications that may occur with each cell division.

Indeed, the functional heterogeneity and limited overlap of differentially expressed genes upon aging of murine HSCs supports the idea that deterioration of HSC functioning during aging may be caused by cumulative and random epigenetically inherited gene expression changes, rather than by mutations in specific loci [69,70,78,79,80,81,82,83,84,85,86,87,88,89],[Lazare et al. 2016(unpublished)]. Although a comprehensive overview of alterations of epigenetic histone and DNA modifications that are present in aged murine HSC is lacking, several age-associated epigenetic alterations have been identified. For example, in old HSCs trimethylation of lysine 4 of histone 3 (H3K4me3) is more widespread, while a gradual increase of global DNA methylation levels is observed. This is accompanied by a reduction of 5-hmC in old HSCs compared to young [79,90]. Concordantly, old HSCs have been reported to down-regulate expression of genes involved in chromatin regulation, such as chromatin remodelling genes (*Smarca4, Smarcb1*), histone deacetylases (*Hdac 1, 5,6*) and DNA methyltransferases (*Dnmt3b*) [69]. Intriguingly, chimeric mice produced from induced pluripotent stem (iPS) cells that were derived from aged HSCs presented with a rejuvenated haematopoietic system, suggesting that it is possible to reprogram aged HSCs and reverse the aging process [91]. These findings strongly suggest that it is indeed—potentially reversible—epigenetic changes that causally contribute to HSC aging.

## 3. Epigenetic Changes during Normal Aging of Human HSCs

Also in the haematopoietic system of humans there are indications that HSC function deteriorates with age. For example, inferior engraftment is seen when HSCs from older donors are transplanted compared to young donors [92]. Beyond differences in functionality, changes in the epigenetic landscape have been observed in aging human HSCs. Several factors are described to be related to these changes, such as loss of fidelity in copying the epigenetic marks during cell divisions, and genetic and environmental factors. Collectively, these lead to epigenetic changes by interfering with regulatory mechanisms [93,94]. The molecular mechanisms and the extent by which these factors contribute to epigenetic aging is still not completely clear. An example of epigenetic changes observed in both aged murine and human HSCs is the reduction in 5hmC levels [79,95].

A most striking observation has been the identification of strongly age-dependent differentially methylated genomic loci in peripheral blood leukocytes of healthy human donors. Indeed, it is now possible to accurately predict the chronological age of an individual based on the differentially methylated status of a very small number of loci in a peripheral blood cell sample [96,97]. Interestingly, this epigenetic age, based on DNA methylation patterns in leukocytes, appears to be cell-intrinsic and independent of the environment, as it does not change upon allogenic haematopoietic stem cell transplantation (HSCT) [98,99]. However, another study that investigated DNA methylation patterns after allogenic HSCT observed that in fact a short period of ‘rejuvenation’ occurs, followed by accelerated aging as measured by DNA methylation levels [100]. Patient groups, conditions, time points and follow-up differed between these studies. This epigenetic clock currently includes only DNA methylation patterns, but may become even more accurate when other age-related epigenetic changes can be included. It remains unclear to what extent this epigenetic clock reflects a history of cell proliferation, and whether it is first established in more primitive haematopoietic cell compartments, including stem cells.

Beyond these age-related epigenetic changes, it has recently become apparent that in otherwise healthy elderly people a significant number of blood cells is derived from one, or only a few, dominant stem cell clones. This is referred to as clonal haematopoiesis of indeterminate potential (CHIP, [101]), or age-related clonal haematopoiesis (ARCH, [102]). Clonal haematopoiesis is characterized by the dominant presence of HSCs that carry somatic mutations in genes that are likely to drive the expansion of a genetically identical clone of haematopoietic cells. Large-scale analysis of blood-derived genetic data from individuals within the Cancer Genomes Atlas who did not display haematological malignancies showed age-related expansion of a haematopoietic stem- or progenitor cell clone and the concurrent presence of leukaemia/lymphoma-associated mutations in about 2% of individuals studied. This frequency increased to 5–6% for individuals who are at least 70 years of age [103]. These intriguing findings have been confirmed in multiple independent cohorts ([103,104,105] and many more). Healthy individuals with clonal haematopoiesis have a significantly higher change to develop haematological malignancies, or indeed other disease [106,107]. Although our molecular understanding of the mechanisms that lead to age-related clonal haematopoiesis at present is far from complete, it is interesting to note that mutations were found in genes encoding for proteins in involved in DNA (de-)methylation (DNMT3A and TET2) and chromatin compaction (ASXL1). 

## 4. Epigenetic Changes in Age-Related Haematopoietic Malignancies

Mutations in these epigenetic modifiers in individuals who display CHIP or ARCH, are not only found in normal aging, but are also considered to be critical in the development of acute myeloid leukaemia (AML) and acute lymphoid leukaemia (ALL) [30]. Presumably, mutations in the aforementioned genes result in aberrant epigenetic regulation in HSCs, which derail proper control of HSC self-renewal programs, and thus constitute the initiating events in the pathogenesis of haematological malignancies. Aberrant epigenetic marks, epi-mutations, are prevalent in many myeloid leukaemias [18]. In leukaemia, (epigenetic) dysregulation may lead to the accumulation of immature, non-functioning pre-malignant cells. The malignant transformation subsequently results from the activation of oncogenes and self-renewal pathways or from silencing of tumour suppressor genes, repression of differentiation or apoptosis pathways. These molecular events can be induced by hypermethylation of promotor regions, increased Polycomb repression or aberrant chromatin conformation [108]. It has been shown in a cohort of 200 cases of de novo AML that 74% of patients had at least one mutation in an epigenetic modifier [109]. The three most recurrently mutated genes in AML affecting these aberrant DNA methylation patterns (*DNMT3A* and *TET2*) and histone modifications (*ASXL1*) will be discussed below.

### 4.1. Aberrant DNA Methylation

DNA methylation patterns have been used to stratify clinically relevant subgroups among AML patient samples [18]. Although clustering appeared possible, patterns were complex and not straightforward to interpret. Whereas ~25% of the differentially methylation clusters corresponded to known AML subtypes, four epigenetic subtypes were defined among patients who shared *NPM1* mutations, some clusters were enriched for specific genetic abnormalities, and novel clusters appeared. Overall, while in most AML cases DNA hypomethylation was prevalent, there were also subgroups that showed hypermethylation [18]. Thus, aberrant hyper- as well as hypomethylation states are found in haematological malignancies.

*DNMT3A*, encoding for a de novo DNA methyltransferase, is one of the most mutated genes in AML. For de novo AML almost 20% of the patients carry mutations in this gene [110], while for de novo MDS this is ~13% [111]. Heterozygous mutations of *DNMT3A* have been observed to lead to the development of myeloid leukaemias, implying that *DNMT3A* qualifies as a haploinsufficient tumour suppressor gene [110,112,113,114]. *DNMT3A* mutations have been associated with poor prognosis (reviewed in [114]), however not in all studies [112]. Functional studies show that loss of the de novo methyltransferase DNMT3A increases HSC self-renewal in mice [8,9,10]. Correspondingly, most *DNMT3A* mutations are loss of function mutations and the most common *DNMT3A* mutation is associated with hypomethylation, although hypermethylated regions have also been found [115,116,117].

*TET2* is another frequently mutated gene in haematopoietic malignancies. *TET2* encodes a protein that catalyzes the conversion of methylcytosine to 5-hydroxymethylcytosine and constitutes the first step towards DNA demethylation. Loss-of-function mutations and deletions of *Tet2* have been associated with DNA hypermethylation [118]. Loss of TET2 increased HSC self-renewal in functional studies in murine HSCs [13,18]. Restoration of TET2 function blocked aberrant self-renewal and leukaemia progression in chimeric mouse models [119]. *TET2* mutations are detected in 7–23% of AML cases, 10–20% of MDS cases [120] and 50% of chronic myelomonocytic leukaemia (CMML) cases [121]. In de novo AML with *TET2* mutations hypermethylated promotor regions were found, overlapping with *IDH1/2* mutated leukaemias [14]. *TET2* mutations are associated with an increased risk of MDS progression and confer a poor prognosis in AML [13,119]. Low *TET2* mRNA levels are related to poor prognosis and *TET2* mRNA levels might be a potential prognostic biomarker for AML since levels increase upon remission and decrease upon relapse [122].

### 4.2. Aberrant Histone Modifications

In addition to changes at the DNA level described earlier, multiple modifications of post-translational histone modifications have been found in haematological malignancies [123]. These aberrant modifications in leukaemia are diverse and often include altered methylation states of histone H3, mainly at lysine residues K4, K9, K27, K36 and K79 [124,125]. Perturbed histone modifications are presumably caused by altered expression of histone modifying enzymes. Indeed, high expression levels of histone deacetylases (*HDAC*) are found in several types of cancer [126]. In myeloid leukaemias the expression levels of *HDAC*s are heterogenous, where *HDAC* levels were generally increased in CLL [127]. Also, overexpression of histone demethylases, such as *LSD1* has been observed in leukaemia [128]. In addition, genes involved in histone methylation and demethylation, for example *ASXL1, EZH2,* and *DOT1L* are frequently mutated in myeloid malignancies.

*ASXL1* is associated with the repressive H3K27me3 mark and is one of the three most commonly mutated epigenetic modifiers genes in AML. *ASXL1* mutations were found to confer a global reduction of H3K27 trimethylation by disrupting normal recruitment of PRC2 [40]. Loss of *Asxl1* was reported to lead to MDS-like disease in mice [41], whereas enforced overexpression of truncated forms of ASXL1 show gain-of-function and promote the pathogenesis of myeloid malignancies [129,130]. Acquired somatic mutations in *ASXL1* occur in approximately 10–20% of patients with MDS and 15–25% in myeloproliferative neoplasms (MPN) and AML [131,132,133,134]. Mutations in *ASXL1* are associated with poor prognosis in malignant myeloid diseases [131,135].

Beyond mutations in epigenetic histone modifiers, mutations affecting lysine residues in histone H3 genes have also been described. These have an inhibitory effect on EZH1/2, thereby causing reductions in H2K27me2/3. These mutations have a high prevalence in paediatric glioblastoma, but were recently also characterized in adult AML representing sporadic mutations and may co-occur with *RUNX1* aberrations [136].

Although these studies clearly suggest a role for aberrant histone modifications in the aetiology of leukaemia, it is difficult to draw definitive conclusions as to how exactly each modification contributes to disease progression. Histone methylation can repress or activate gene expression, depending on which lysine is methylated. For definitive molecular insight it would be required to obtain single cell, single locus and single molecule resolution of histone modifications, and current technology does not permit such approaches. Additionally, discerning between levels of active and repressive marks is complicated by the fact that leukaemic mutations in methyltransferases that deposit active marks, such as the H3K4 and H3K36 methyltransferases, are both gain and loss of function, respectively [125]. Most mutations in genes encoding PRC2 proteins, involved in deposition of the repressive H3K27me3 mark, such as *SUZ12, EED* and *EZH2*, , are loss of function mutations. In contrast, mutations in genes encoding for proteins involved in the deposition of other repressive marks, such as H3K9me1 methyltransferases, turn out to be gain of function mutations [125]. Similar as the DNA hypo- and hypermethylated states that are found in AML (as described above), both gain and loss of H3K27me3 levels seems to promote malignant transformation. Similarly, both overexpression and loss of *EZH2* are associated with malignant transformation [23,137]. Loss-of-function mutations have been described in CMML and in AML [137,138,139], whereas gain-of-function mutations are frequently encountered in lymphoma patients [140], but also in chronic myeloid leukaemia (CML) and lymphoid leukaemias [137]. In case of MDS and T-ALL, both overexpression and inactivating mutations have been described [137], indicating aberrant expression in either way may drive oncogenesis. It is conceivable therefore that any perturbation of the stochiometry of epigenetic writers, readers and erasers in large protein complexes may cause primitive haematopoietic cells to proliferate uncontrollably and, thus, results in malignancy.

## 5. Epigenome-Targeted Therapies

The presence of epigenetic modifications in leukaemia and the potential reversible nature of many epigenetic modifications suggest that compounds that affect the epigenetic machinery may be used for therapeutic purposes. Indeed, multiple pharmacological intervention strategies to reverse potentially pathological epigenetic modifications are in development (Table 2). DNA methyltransferase inhibitors (DNMTi) were the first FDA approved drugs targeting the epigenome, aimed to treat haematological diseases, such as MDS, AML and CMML [141]. In addition to drugs that affect DNA methylation, the clinical potential of compounds that target proteins involved in histone modifications is also actively explored. The first histone deacetylase inhibitors (HDACi), approved for multiple myeloma and cutaneous T cell lymphoma, are in clinical trials [142]. Histone methyltransferase inhibitors (HMTi) and histone demethylase inhibitors (HDMi) are emerging as strategies in epigenetic-based pharmacology. Small-molecule inhibitors of histone acetyltransferase (HATi) are investigated in pre-clinical studies. These approved DNMTi and HDACi, initially used for the treatment of haematological malignancies, are currently also being tested in clinical trials for other diseases, such as neurological diseases, latent viral infections and immune diseases [143]. 

### 5.1. DNA Methylation Inhibitors (DNMTi)

DNMTi are nucleoside analogues that form a covalent complex with DNA methyltransferases (DNMT) and thereby inhibit ongoing methylation. After their incorporation into DNA, these inhibitors sequester DNMT enzymes, leading to gradual and global demethylation upon cell division. The two most commonly used DNA methylation inhibitors, azacitidine and decitabine, have been evaluated in several clinical trials and are incorporated in clinical guidelines [141]. Azacitidine is approved for MDS and AML and provides an increase in overall survival in older AML patients with poor prognostic karyotypes and high risk MDS (AZA001 trial) [144]. However, it is important to note that only half of the treated patients show a response to azacitidine treatment. Criteria to stratify patients who will respond to these drugs have not yet been identified [145]. Overall survival of MDS patients in the AZA001 trial was superior to survival in cohorts of patients treated in normal clinical practice outside of clinical trials, discouraging routine treatment with azacitidine above enrolment in clinical trials [146]. Survival rates of patients with AML and MDS with cytogenetic abnormalities associated with unfavourable risk, with *TP53* mutations, or with both, were similar to patients with an intermediate-risk cytogenetic profile after decitabine treatment [147]. This suggests a place for decitabine in treating these patients. In addition, these DNA hypomethylating agents were reported to have higher overall response rates in MDS and AML patients with *TET2* mutations, associated with DNA hypermethylation [118]. Counterintuitively, hypomethylating agents also appear effective in *DNMT3A* mutated AML, which is associated with hypomethylation, and a small study suggests patients might respond even better [148]. These hypomethylating agents may target specific hypermethylated regions or may result in even further hypomethylation of the genome [114].

The disadvantage of DNMTi is that they induce genome-wide hypomethylation and lack locus specificity, which could lead to activation of oncogenes and/or increased genomic instability. Further research into the long term effects of treatment with DNMTi might learn us more about the therapy related effects and whether the association with therapy related malignancies is comparable to that of commonly used chemotherapeutics. Considering the heterogeneity of AML as a disease, the effectiveness of DNMTi will likely depend on the leukaemia subtype and is likely to vary from patient to patient. Insight into the molecular mechanism of aberrant DNA methylation patterns and the following development of criteria for responders can lead the way to use of these drugs for the right subgroups of AML patients. It is relevant to note that upon azanucleoside withdrawal DNA methylation levels return to pre-treatment levels [149]: demonstrating a continuous need for DNA methyltransferase inhibition. 

### 5.2. Histone Deacetylase Inhibitors (HDACi) and Histone Acetyltransferase (HATi)

Histone acetylation is an epigenetic mark that is typically associated with active gene expression. Concordantly, inhibition of histone deacetylases by HDACi has been shown to derepress expression of genes involved in cell death, growth arrest and differentiation in myeloid malignancies [142,150,151,152,153]. The mechanism of action of most HDACi is by binding to the catalytic site of the HDAC enzyme. HDACi can also affect acetylation of non-histone proteins potentially leading to more general effects [125], including the inhibition of DNA repair processes [154]. For example, DNA damage can be induced by the HDACi vorinostat, possibly by creating a more susceptible, open chromatin conformation. There are reports that suggest that HDACi-induced DNA damage can be repaired by normal cells but not by malignant cells [153]. Treatment of AML or MDS patients with HDACi only, showed low response rates in 10–20% [155].

The use of different combinations of drugs with HDACi is currently being explored. Cameron et al. have addressed the rationale for such combination, e.g., HDACi and DNMTi already in 1999 [156]. An approved HDACi combination is panobinostat combined with bortezomib and dexamethasone for treatment of multiple myeloma (FDA and EMA approved in 2015) [157]. Sequential administration of DNMTi and HDACi demonstrated clinical efficacy in patients with haematologic malignancies [158]. Additionally, pracinostat (SB939) combined with azacitidine is tested for efficacy and safety in a phase III AML trial (NCT03151408) [143]. Beyond inhibition of histone deacetylases, inhibition of histone acetyltransferases has also been evaluated [159]. This includes inhibition of p300 with small molecules to induce apoptosis in AML cells in pre-clinical setting [160]. 

### 5.3. Histone Methyltransferase Inhibitors (HMTi) and Histone Demethylase Inhibitors (HDMi) 

Disruption of histone methylation patterns is another potential therapeutic strategy to (de)repress differentiation-inducing genes in cancer. The histone methyltransferase inhibitor tazemetostat, an EZH2 inhibitor, is tested for relapsed or refractory B cell NHL with *EZH2* mutations [161,162]. Inhibitors targeting histone demethylases, such as the lysine specific demethylase (LSD1) inhibitor, INCB059872, are evaluated in phase I/II clinical MDS/AML trials [143].

### 5.4. Inhibiting Epigenetic Readers

Broadly acting epigenetic drugs will lead to genome wide alterations of epigenetic modifications. DNMTi, for example, have genome-wide effects on the DNA methylome, leading to acute genome-wide DNA demethylation of repetitive elements, as well as loss of methylation of specific tumour suppressor genes and normalization of cell growth [143]. To reduce these genome wide effects and associated side effects, more specific epigenetic targeting therapies are investigated in preclinical settings. Small molecules that specifically target the binding of epigenetic modifier proteins may be of use for this. The development of BET inhibitors, that preferentially prevent binding of bromodomain-containing epigenetic reader proteins to acetylated histones and transcription factors, served as a proof of principle that there are epigenetic drugs which are able to inhibit certain protein interactions more specifically [123,172]. Development of epigenetic drugs directed against specific protein-protein interactions and thereby influencing epigenetic regulation is ongoing, including compounds that prevent binding of PRC1/2 complexes (EZH2—[162], CBX7—[47,173,174,175]).

## 6. Future Prospects

It is important to remember that our knowledge of molecular mechanisms that contribute to HSC aging is essentially based on studies in mice, whereas the clinical implications (anaemia, increased susceptibility to infections and malignancies) are evident in human studies. We assume that the clinical observations made in human patients result from the molecular changes that have been well documented in mice, but at this point this remains to be proven. To cover this gap in knowledge, more insight in human HSC aging and robust mouse aging models are required. This also relates to the concept of age-related clonal haematopoiesis, which is very well described in humans, but much less in mice.

As less than one percent of elderly people with clonal haematopoiesis develop leukaemia, identification of those who will eventually develop haematological malignancies is of great interest. Once these individuals can be accurately identified it may be possible to prevent development of leukaemia by specifically targeting the emerging preleukaemic clones in elderly people. For reasons that we do not understand, elderly people with age-related clonal haematopoiesis also have increased risk to present with a wide variety of non-haematological conditions [106,107]. It is tempting to speculate that eradicating aberrant haematopoietic stem cell clones may also be beneficial to prevent these non-haematological diseases.

Once persistent and systematic epigenetic changes have been identified, the plasticity of such lesions renders them an interesting target for novel therapeutic strategies. Epigenetic therapy in AML is in development, many new targets and inhibitors are being explored. The first epigenetic compounds are already approved for clinical use in haematopoietic malignancies and more specific epigenetic therapies are in development and are studied in (pre)clinical studies. Finding ways to specifically target the aberrant epigenetic marks in abnormal HSCs while sparing healthy HSCs would potentially be of significant clinical interest. These newly developed epigenetic drugs may be beneficial as single agent, but also in combinations with commonly used chemotherapeutics, immune therapy or other epigenetic drugs [176]. Reactivation of epigenetically silenced apoptotic genes may increase efficacy of chemotherapy.

Unfortunately, development of new drugs for AML is impeded by the heterogeneity of this disease. The resulting variability in response rates warrants the need for disease stratification and biomarkers to discern which patients will benefit from new treatments. Epigenetic aberrations and mutations in epigenetic modifiers may be useful in this context [177,178,179]. Beyond patient stratification, epigenetic alterations may play a role in prognostic evaluations since mutations in several epigenetic modifiers are associated with clinical outcome in patients with malignant diseases (for example *DNMT3A, TET2, IDH2, MLL, EZH2* and *ASXL1*; reviewed in [123]).

Upon aging many changes in the epigenetic landscape of HSCs occur, both in the murine and the human haematopoietic system. However, the functional consequences of these epigenetic perturbations, although likely to be relevant, remain unclear at this point. Elucidating the functional role of epigenetic modifications that occur during aging and during malignant transformation is partially provided by large-scale analysis of epigenetic modifications in patients with haematological disease (such as the BLUEPRINT project, part of The International Human Epigenome Consortium) [180]. Such important descriptive studies must be followed by functional studies unravelling how epigenetic changes are regulated and lead to aberrant gene expression and phenotypes.

## Figures and Tables

**Figure 1 cells-08-00868-f001:**
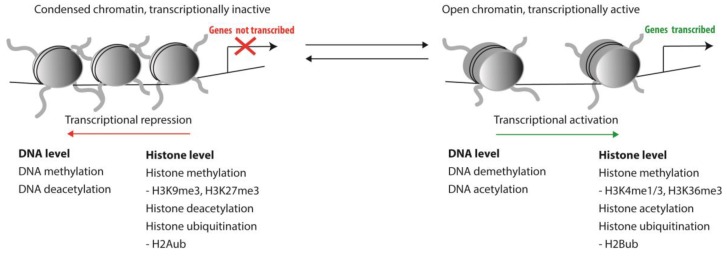
Epigenetic modifications and their effect on DNA accessibility.

**Table 1 cells-08-00868-t001:** Overview of major epigenetic writers, readers and erasers that play a role in HSC self-renewal.

Gene	Function	Effect on HSC	Reference
**DNA modifications**			
*DNMT1*	DNA methyltransferase: maintenance parental cell methylation patterns.	Required for HSC self-renewal, niche retention and progression from multipotent to myeloid progenitors. Deletion leads to lineage skewing towards myelopoiesis and defective self-renewal.	[4,5,6]
*DNMT3A/B*	DNA methyltransferase: de novo DNA methylation.	Essential for HSC self-renewal and deletion increases HSC life span. DNMT3A and DNMT3B show complementary de novo methylation patterns responsible for silencing of self-renewal genes in HSCs.	[7,8,9,10,11]
*TET1/2*	Catalyse the oxidation of 5-methylcytosine into 5-hydroxymethylcytosine, resulting in DNA demethylation.	TET1 deficiency increases HSC self-renewal potential. *TET2* deletion resulted in enhanced HSC self-renewal and enhanced myelopoiesis. *TET2* mutations are mutually exclusive with gain of function mutations *IDH1/2* in AML.	[12,13,14,15,16,17]
*IDH1/2*	Isocitrate dehydrogenase 1/2 enzymes (IDH1/2), required for conversion of isocitrate into a-ketoglutarate, a TET2 cofactor.	*IDH1* mutations impair histone demethylation and result in hypermethylation. TET2 function becomes disrupted and differentiation is inhibited. Mutant *IDH1/2* overexpression in primary bone marrow cells disturbs haematopoietic differentiation and leads to elevated numbers of HSPCs.	[18,19,20]
**Histone modifications**			
*EZH1/2*	Histone lysine methyltransferase, PRC2 member.	Important for HSC maintenance. Deletion decreases self-renewal potential. *EZH2* deletion alone does not compromise self-renewal, possibly due to complementary EZH1 activity.	[21,22,23,24]
*DOT1L*	H3K79 methyltransferase	Crucial for embryonic erythropoiesis. Not essential for adult haematopoiesis.	[25,26,27,28]
MLL proteins	H3Kmethyltransferases	MLL-fusion enriched target genes play a role in HSC function, MLL mediated activation of *HOX* genes has been described, MLL fusions are potent inducers of leukaemia.	[29,30,31,32]
*SETDB1*	H3K9 methyltransferase	Essential for HSC function.	[33]
*PMRT4/5*	Protein arginine methyltransferases	PMRT4 blocks myeloid differentiation. PMRT5 inhibits expression of differentiation associated genes, *PMRT5* deletion results in HSPC exhaustion.	[34,35,36,37]
*CBP/p300 MOZ*	Histone acetyltransferases	CBP/p300 regulates self-renewal and differentiation in adult HSCs. MOZ maintains the generation and development of HSCs.	[38,39]
*ASXL1*	Polycomb chromatin-binding protein, associates with PRC1 and PRC2.	Loss results in impaired self-renewal.	[40,41,42]
*CBX2*	Chromobox 2, PRC1 member. Reads H3K27me3.	Impairs HSC and progenitor proliferation.	[43,44,45]
*CBX7*	Chromobox 7, PRC1 member. Reads H3K27me3 and other trimethylated proteins.	Overexpression increases HSC self-renewal.	[45,46,47]
*Ring1B*	E3 ubiquitin-protein ligase.	Antiproliferative role in progenitor expansion.	[48]
*BMI1/MEL-18*	Polycomb ring finger protein, PRC1 member.	Repression impairs self-renewal. Frequently overexpressed in malignant haematopoiesis. Loss enhances HSC self-renewal.	[49,50,51,52]
*LSD1*	H3K4/9 demethylase	Overexpression of the short isoform increases self-renewal potential. Loss causes defects in long-term HSC self-renewal and impaired differentiation.	[53,54,55]
*JARID1b*	H3K4 demethylase	Described as a positive regulator of HSC self-renewal capacity.	[56,57]
*KDM6A/UTX*	H3K27 demethylase	Knock out leads to myelodysplasia, suppressed megakaryocytopoiesis and extramedullary haematopoiesis, indicating a regulatory role in haematopoiesis.	[58,59]
*KDM6B/JMJD3*	H3K27 demethylase	Loss impairs HSC self-renewal potential following proliferative stress.	[59,60]
*MYSM1*	Histone H2A deubiquitinase	Maintenance of HSC function. Deletion results in impaired self-renewal.	[61,62,63]
*SIRT1/2/3/7*	NAD-dependent deacetylases	*Sirt1* loss increases HSC numbers. *Sirt2* and *3* are not required for HSC maintenance at young age, but knockout does compromise HSC function at old age. *Sirt6*-deficient HSCs exhibited impaired self-renewal ability. *Sirt7* overexpression increases constitutive capacity of HSCs.	[64,65,66,67,68]

**Table 2 cells-08-00868-t002:** Examples of epigenetic drugs and their stage of development for treatment of haematological malignancies.

Type of Drug	Target Enzyme	Compound	Stage of Development	References
DNMTi	DNA methyltransferase	Azacitidine, decitabine	EMA and FDA approval for MDS, AML and CMML	[163]
		IDH1/2 inhibitors	Pre-clinical and clinical	[164,165,166]
		Ivosidenib (IDH1)	EMA and FDA approval for *IDH1* mutated and R/R AML	
		Enasidenib (IDH2)	EMA and FDA approval for R/R AML	
HDACi	Histone deacetylases	Panobinostat	EMA and FDA approval for MM, CTCL	[125,142,152,154]
		Romidepsin	EMA and FDA approval for CTCL	
HATi	Histone acetyltransferases	P300 inhibitors	Pre-clinical	[159,160,167]
HDMi	Histone demethylases	LSD1 inhibitors	Pre-clinical and clinical	[128,168]
HMTi	Histone methyltransferases	EZH1/2 inhibitors Tazemetostat (EZH2)	Pre-clinical and clinical	[169,170]
		DOT1L inhibitors	Pre-clinical and clinical	[128,171]
		MENIN1-MLL interaction inhibitors	Pre-clinical	[128]
HATi	Histone acetyl readers	BET-inhibitors	Pre-clinical and clinical	[163,166]
(myelodysplastic syndrome (MDS), relapsed/refractory (R/R) acute myeloid leukaemia (AML), chronic myelomonocytic leukaemia (CMML), multiple myeloma (MM), cutaneous T cell lymphoma (CTCL))				

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
