# Peer review of "Epigenetic Changes as a Target in Aging Haematopoietic Stem Cells and Age-Related Malignancies"

_cells, 2019, doi:10.3390/cells8080868_

Round 1

Reviewer 1 Report

Buisman and De Haan reviewed the recent studies on epigenetic regulation of hematopoietic stem cell aging. This is a rapidly moving field of important clinical significance. The authors summarized the compelling evidence on genetic and biochemical evidence supporting epigenetic regulation of hematopoietic stem cell aging and malignancies in mouse models and human samples. The authors also reviewed the state of the art of therapeutic development targeting the epigenome.  This is an important contribution to the field. The manuscript is nicely written. Viewpoints are well balanced. I support its publication.

Minor Comment:

SIRT2 was recently shown to regulate HSC aging (Luo, Cell Reports 2019), expanding the sirtuin family in such a regulation.

Author Response

We like to thank the reviewers for their constructive critiques on our manuscript, our response to the addressed comments is provided point-by-point below (in red).  

Minor Comment:
SIRT2 was recently shown to regulate HSC aging (Luo, Cell Reports 2019), expanding the sirtuin family in such a regulation.

The information about the proteins of the sirtuin family was expanded in table 1.

Reviewer 2 Report

Epigenetic alterations are associated with hematopoietic stem cell (HSC) aging and hematopoietic malignancies. Thus the topic of this review article is significant and timely in light of recent efforts in understanding the epigenetic mechanisms that control normal HSC aging and age-related hematopoietic diseases. This review provides an updated summary of the current knowledge of epigenetic regulation/dysregulation in age-associated malignancies, and many examples of how alterations of epigenetics underlie disease phenotypes. The author also discussed updates on current efforts of developing epigenetic drugs as potential treatment options for these diseases.  Overall, this is a well written review addressing important questions related to epigenetic regulation of HSC aging and diseases. This reviewer had a few minor suggestions to further improve this review:

Figure 1, it is unclear what the color arrowheads mean in the figure. The authors should describe clearly in the figure legends or use another means of illustration.

Table 1, it would be helpful to also include histone demethylases (e.g. UTX and JMJD3) in the section of erasers given recent reports on these factors in HSCs and hematopoietic diseases.

Section 4.2, it would be helpful to also discuss EZH2 gain-of-function mutations in lymphomas and include relevant citations as additional examples of the context-specific roles of EZH2 in malignancies.

Section 4.2, also it would be helpful to discuss mutations associated with histones (e.g. H3K27M mutation) as additional examples of epigenetic alterations that are recently described in solid tumors and blood cancers.

A few typos: line 152, CMLL should be CML, etc.

Author Response

We like to thank the reviewers for their constructive critiques on our manuscript, our response to the addressed comments is provided point-by-point below (in red).  

Figure 1, it is unclear what the color arrowheads mean in the figure. The authors should describe clearly in the figure legends or use another means of illustration.

We were happy for this constructive remark, figure 1 was edited to clarify the meaning of the figure.

Table 1, it would be helpful to also include histone demethylases (e.g. UTX and JMJD3) in the section of erasers given recent reports on these factors in HSCs and hematopoietic diseases.

Information about the H3K27 histone demethylases is now included to table 1.

Section 4.2, it would be helpful to also discuss EZH2 gain-of-function mutations in lymphomas and include relevant citations as additional examples of the context-specific roles of EZH2 in malignancies.

We agree with the reviewer this is helpful to mention. The part in section 4.2 about EZH2 mutations was expanded, more references were used to support the statement that EZH2 aberrations in either way are associated with hematological malignancies.

Section 4.2, also it would be helpful to discuss mutations associated with histones (e.g. H3K27M mutation) as additional examples of epigenetic alterations that are recently described in solid tumors and blood cancers.

It is indeed interesting to mention that not only mutations in epigenetic modifiers, but also in mutations associated with histones are found in hematological malignancies. The fact that mutations that can occur in histones, with H3K27M in AML as an example, is now mentioned in section 4.2.

A few typos: line 152, CMLL should be CML, etc.

Line 152: CMLL changed into chronic myelomonocytic leukemia (CMML) The manuscript was evaluated for other typo’s.

Reviewer 3 Report

This review summarises epigenetic regulation of hematopoietic stem cells and changes in the epigenome in mice and humans with increased age. It then goes on to summarise epigenetic changes related to leukaemias or pre-neoplastic disease to finally discuss epigenetic therapy in this context. This review is well written, comprehensive and covers all aspects mentioned in the title.

I have a few minor pints I would like the authors to address.

The authors touch on the relationship between proliferation and epigenetic changes. I would like this aspect to be extended on with the view that there is active and passive demethylation of the genome, with passive demethylation being linked to DNA replication. Lines 88 onwards: The presented literature is somewhat incomplete see doi: 10.3324/haematol.2016.160481 I think the extent of the environment on epigenetic age is currently not very clear cut and merits further discussion. I also find the correlation between epigenetic age and histone modifications extremely speculative. Paragraph on Epigenome-targeted therapies

a.) When DNA methylation inhibitors are used in patients, they can lead to activation of oncogenes and/or increased genomic instability.  That means they are not reliable enough/ safe for first-line treatment in patients. In the clinic, DNMTi like decitabine is used either in combination with chemotherapy or given in a small dose.  The combination of DNMTi with chemo showed an increased effect compared to chemotherapy.  This point should be discussed more thoroughly

b) It has been shown that BET inhibitors engaged with the immune system to kill off cancer cells in heamatological maliganancies (Lymphoma, CML, https://doi.org/10.1016/j.celrep.2017.02.011). This pathway requires and intact host-immune system. This research suggests a novel approach using immune system activation to target tumor cells. It would be nice to compares between direct epigenetic targets and newly identified immune system targets to, in the future, investigate which one is more effective. 

Author Response

We like to thank the reviewers for their constructive critiques on our manuscript, our response to the addressed comments is provided point-by-point below (in red).  

The authors touch on the relationship between proliferation and epigenetic changes. I would like this aspect to be extended on with the view that there is active and passive demethylation of the genome, with passive demethylation being linked to DNA replication.

We agree with the reviewer it is interesting to mention that not only active but also passive demethylation takes place, next to the epigenetic changes that occur during cell division. A paragraph about epigenetic changes during cell division, including passive demethylation, has been added.

Lines 88 onwards: The presented literature is somewhat incomplete see doi: 10.3324/haematol.2016.160481

We addressed a part of the know literature regarding this subject, we now present a more comprehensive overview of related literature with among others the article mentioned by the reviewer.

I think the extent of the environment on epigenetic age is currently not very clear cut and merits further discussion.

Factors related to epigenetic aging, such as the environment, are now mentioned.

 I also find the correlation between epigenetic age and histone modifications extremely speculative.

We would like to thank the reviewer for this comment, our idea was not to mention a correlation, we tried to imply that if the basis for the epigenetic clock would be expanded with other epigenetic marks than merely the DNA methylation status it might become even more precise. We have rewritten the sentence in a way that fits better to our intention and is less speculative.

Paragraph on Epigenome-targeted therapies

a.) When DNA methylation inhibitors are used in patients, they can lead to activation of oncogenes and/or increased genomic instability.  That means they are not reliable enough/ safe for first-line treatment in patients. In the clinic, DNMTi like decitabine is used either in combination with chemotherapy or given in a small dose.  The combination of DNMTi with chemo showed an increased effect compared to chemotherapy.  This point should be discussed more thoroughly

This in an important point to address, since it should be clear that many commonly used cancer therapeutics have a lot of direct side effects and toxicity, next to long term, potentially pro-oncogenic, effects. Many chemotherapeutics are associated with an increased cancer risk. One of the examples of this is the association of chemotherapy for solid tumors with therapy related hematological malignancies, recently published in JAMA oncology(DOI:10.1001/jamaoncol.2018.5625). The fact that the DNMTi are already in the clinics does not warrant they do not have of target effects. To clarify this point we added a sentence about how future research can learn us more about the long term effects of DNMTi and whether the association with therapy related malignancies is comparable to commonly used chemotherapeutics.

b) It has been shown that BET inhibitors engaged with the immune system to kill off cancer cells in heamatological maliganancies (Lymphoma, CML, https://doi.org/10.1016/j.celrep.2017.02.011). This pathway requires and intact host-immune system. This research suggests a novel approach using immune system activation to target tumor cells. It would be nice to compares between direct epigenetic targets and newly identified immune system targets to, in the future, investigate which one is more effective. 

Indeed the studies on BET inhibitors engaging with the immune system are covering a very interesting field of research. The main focus of our review is to give an overview of epigenetic changes, targets and epigenetic drugs. Already a broad focus which forced us to leave further associations outside of the scope of this review and made it impossible to cite all relevant work. To make a proper and well-founded comparison about direct epigenetic targets and immune system targets with the related background information would in our opinion distract from the main focus of the review and would not fit the manuscript in the form it is now.